# Non-Coding RNAs, a Novel Paradigm for the Management of Gastrointestinal Stromal Tumors

**DOI:** 10.3390/ijms21186975

**Published:** 2020-09-22

**Authors:** Azadeh Amirnasr, Stefan Sleijfer, Erik A. C. Wiemer

**Affiliations:** Department of Medical Oncology, Erasmus MC Cancer Institute, Erasmus University Medical Center, 3015 CN Rotterdam, The Netherlands; a.amirnasr@erasmusmc.nl (A.A.); s.sleijfer@erasmusmc.nl (S.S.)

**Keywords:** microRNA, long non-coding RNAs, GIST, biomarker, therapy

## Abstract

Gastrointestinal stromal tumors (GISTs) are the most common mesenchymal malignancies found in the gastrointestinal tract. At a molecular level, most GISTs are characterized by gain-of-function mutations in V-Kit Hardy–Zuckerman 4 Feline Sarcoma Viral Oncogene Homolog (*KIT*) and Platelet Derived Growth Factor Receptor Alpha (*PDGFRA*), leading to constitutive activated signaling through these receptor tyrosine kinases, which drive GIST pathogenesis. In addition to surgery, treatment with the tyrosine kinase inhibitor imatinib forms the mainstay of GIST treatment, particularly in the advanced setting. Nevertheless, the majority of GISTs develop imatinib resistance. Biomarkers that indicate metastasis, drug resistance and disease progression early on could be of great clinical value. Likewise, novel treatment strategies that overcome resistance mechanisms are equally needed. Non-coding RNAs, particularly microRNAs, can be employed as diagnostic, prognostic or predictive biomarkers and have therapeutic potential. Here we review which non-coding RNAs are deregulated in GISTs, whether they can be linked to specific clinicopathological features and discuss how they can be used to improve the clinical management of GISTs.

## 1. Gastrointestinal Stromal Tumors: A Brief Introduction

Gastrointestinal stromal tumors (GISTs) are rare tumors of mesenchymal origin from the gastrointestinal (GI) tract with an estimated annual incidence of 10–20 per 1,000,000 in the population [1,2,3]. They can be found anywhere along the GI tract, but occur most commonly in the stomach (60–70%) and small intestine (20–30%) [4]. GISTs are believed to originate from the interstitial cells of Cajal (ICC) or their precursor cells [5,6]. In the GI tract, ICC operate as pacemaker cells responsible for peristaltic movement. The fact of GIST originating from ICC is exemplified by shared immune-phenotypical features such as the expression of V-Kit Hardy–Zuckerman 4 Feline Sarcoma Viral Oncogene Homolog (*KIT*) (CD117) [5,6], anoctamin 1 (ANO1/DOG1) [7] and ETV1 [8], which are currently used as diagnostic biomarkers for GIST. Activating mutations in KIT or Platelet Derived Growth Factor Receptor Alpha (*PDGFRA*) were identified as oncogenic drivers in GIST [9,10,11] (Figure 1A). The gain-of-function mutations in these receptor tyrosine kinases are mutually exclusive and cause constitutive kinase activity in the absence of growth factor binding. Activated KIT and PDGFRA signaling stimulate downstream pathways such as the RAS–RAF–MAPK, PI3K–AKT and JAK/STAT pathways inhibiting apoptosis and promoting cellular survival and proliferation [12] (Figure 1B).

Approximately 80% of GISTs contain a mutation in *KIT* at specific locations in exon 11 (90%), exon 9 (8%) or—less often—in exon 13 (1%) or exon 17 (1%). *KIT* exon 11 encodes the juxtamembrane region, and mutations in this protein domain impair the autoinhibitory activity of the receptor. The mutations detected in exon 9 are supposed to imitate the conformational changes following ligand binding leading to receptor dimerization and activated signaling. Mutations in exon 13 act on the ATP-binding region of *KIT* while mutations in exon 17, which codes for the activation loop of the kinase, stabilize the receptor in its active conformation. *PDGFRA* mutations occur in 10–15% of GISTs, most commonly in exons 12, 14 or 18. The specific mutations in *KIT* and *PDGFRA*, with the exception of PDGFRA D842V, make GISTs amenable to treatment with the tyrosine kinase inhibitor imatinib. This drug selectively inhibits the kinase activity of KIT and PDGFRA through competitive binding at the ATP binding site of these enzymes [13,14,15]. In a minority of GIST cases (5–10%) no mutations in *KIT* or *PDGFRA* can be detected. In these so-called wild-type GISTs (WT-GIST), other mutated genes such as *NF1*, *BRAF* and succinate dehydrogenase subunits (*SDHB*, *SDHC*, *SDHD*) can drive tumorigenesis [16,17,18,19,20].

## 2. Current Treatment of Gastrointestinal Stromal Tumors

Surgical excision is the preferred treatment modality for localized GIST aiming for resection margins devoid of tumor cells [21]. Prior to surgery, imatinib may be administered if complete resection is difficult without downsizing the tumor. If a routine risk assessment, which is usually based on parameters such as the mitotic rate, tumor size and tumor location, indicates a significant chance of relapse after surgery, adjuvant imatinib may be prescribed for up to 3 years [22]. The efficacy of imatinib treatment may vary and is partly dependent on the *KIT* or *PDGFRA* mutational status of the tumor [23]. For example, GISTs that harbor *KIT* exon 11 mutations generally respond well to imatinib [24,25], whereas patients with an exon 9 *KIT* mutation frequently need an increased daily dose of 800 mg/day instead of the regular 400 mg/day to exhibit a treatment response [26]. Furthermore, PDGFRA D842V mutants are resistant to imatinib [27,28], the same as WT-GISTs and GISTs with mutations in genes other than *KIT* and *PDGFRA* that display insensitivity to imatinib and other tyrosine kinase inhibitors [29].

Imatinib is listed as first-line treatment for locally advanced, unresectable and metastatic GIST. In this context, imatinib is usually prescribed indefinitely as pausing treatment generally leads to tumor progression [21]. Unfortunately, the vast majority of GIST patients treated with imatinib eventually present with tumor progression due to the development of drug resistance [23]. The precise molecular changes and mechanisms underlying imatinib resistance are not completely clear. In about half of patients, secondary mutations arise in *KIT*, normally in exon 13, 14, 17 or 18, that cause resistance [30,31,32]. In the remaining half of resistant patients, other less-defined resistance mechanisms are operational [30,33,34,35,36,37]. The standard second-line treatment is currently sunitinib [38,39] with regorafenib as the third line option [40].

Other tyrosine kinase inhibitors are under development of which avapritinib and ripretinib are the most promising and tested in phase III studies [41,42]. These drugs were shown to have inhibitory activity in advanced GISTs resistant to approved treatments and in GISTs with a PDGFRA D842V mutation.

## 3. Clinical Needs Regarding the Management of Gastrointestinal Stromal Tumors

GIST is routinely diagnosed based on specific morphological features, immunostaining for KIT and ANO1 (DOG1) and the presence of *KIT* or *PDGFRA* mutations. A risk assessment is made by the pathologist based on the mitotic count observed in a tumor biopsy, by assessing the tumor size and tumor location. Additional biomarkers that can be quantitatively measured in a standardized fashion may be very useful to further fine-tune the grading procedure. Additionally, one can think of biomarkers that highlight metastasis and can be determined in the patient’s tumor and/or circulation. Although effective treatments exist for GIST, most notably imatinib, almost all patients ultimately develop resistance. Biomarkers that indicate the development of resistance may not only provide insight into the specific mechanisms of resistance, leading to the development of novel strategies to overcome resistance, but also enable the clinician to adjust treatment before overt progression occurs. Last but not least, novel therapeutic approaches are needed that target the oncogenic pathways in GIST differently than the tyrosine kinase inhibitors, giving rise to lasting responses while circumventing resistance.

## 4. Non-Coding RNAs

Novel classes of RNA transcripts, including microRNAs and long non-coding RNAs, have recently been discovered in eukaryotic cells. Their tissue-specific expression, role in gene regulation and their intricate, often essential, involvement with normal and pathological physiology makes them particularly suitable as biomarkers and endows them with therapeutic potential.

The sequencing of the human genome initially indicated the presence of approximately 30,000 protein coding genes [43,44], a number that over the years was adjusted to about 20,500 protein coding genes [45]. GENCODE (www.gencodegenes.org) lists, in its most recent version (release 35), 19,954 protein coding genes. This number of genes is comparable to that found in other—quite often less complex—organisms [46], implicating that organismal complexity is not determined by protein coding gene numbers alone. In fact, the protein coding genes constitute only 1.5% of the human genome but, intriguingly, about two-thirds of the genome is transcribed into RNA [47,48,49]. This vast transcriptional output cannot be all considered as transcriptional noise as that would be an utter waste of cellular energy. Based on these facts it is proposed that organismal complexity is driven by the expansion of the regulatory potential of the non-coding portion of the genome [50]. There is growing evidence that non-coding transcripts exercise diverse biological functions that are still ill-defined or, more often, not yet assigned in most cases. Several classes of RNA transcripts have been recognized and a start has been made to functionally annotate these biomolecules. This review will focus on the rather well-defined subset of microRNAs (miRNAs), small regulatory RNAs of 19–26 nucleotides, and briefly touch upon long non-coding RNAs (lncRNAs) and their subclass circular RNAs (circRNAs) in the context of GIST. miRNAs were first described in the mid-nineties of the last century in the nematode *Caenorhabditis elegans* [51]. Initially, miRNAs were considered a peculiarity of these worms until it was realized that many miRNAs are evolutionarily conserved suggesting a functional relevance for miRNAs [52,53]. Currently, there are 2654 mature human miRNAs listed in miRBase (version 22.1; www.mirbase.org/) and it is well established that miRNAs play pivotal roles by regulating many fundamental developmental and cellular processes [54]. Although exceptions have been reported [55,56] miRNAs most commonly operate by binding in the context of the RNA-induced silencing complex (RISC) to the 3′ untranslated region (3′ UTR) of target mRNAs. The miRNA–mRNA interaction in the context of the RISC causes translation inhibition and/or mRNA degradation. In this way miRNAs are capable of regulating gene expression. Interestingly, any given miRNA may target multiple mRNAs and, conversely, a single mRNA can be targeted by multiple miRNAs. In this way a refined regulatory network is created which itself again can be modulated in various ways and at different levels. It is estimated that two-thirds of all genes are under regulation by miRNAs [57,58], and by inference it is safe to state that miRNAs are small riboregulators involved in almost all—if not all—biochemical and cellular processes. Just as miRNAs are intimately related to normal cellular, tissue and organismal physiology, they also play essential roles in diseases including cancer [59,60,61].

A common feature of cancer is the dysregulation of miRNA expression caused by genomic alterations, amplification and deletions, that are frequently encountered [62]. Alternatively, epigenetic mechanisms may underlie the aberrant expression of miRNAs. It is well established that miRNAs can carry out essential oncogenic and tumor-suppressive roles in the tumorigenic process. Additionally, miRNAs are also known to play a driving role in metastasis [63] and drug resistance [64], thereby affecting the outcome of drug treatment. The close involvement of miRNAs with many biological and clinical aspects of cancer, their tissue-specific expression and quantitative detection methods define miRNAs as suitable biomarkers. An advantage in this respect is that miRNAs are stable and present in many tissues and body fluids such as urine, saliva and blood [65,66]. Driven by academic progress that highlights the key roles miRNAs play in all kinds of disorders, the pharmaceutical industry and biotech developed an interest in miRNA-based therapeutics. Despite significant initial technical challenges related to safety, stability and delivery, numerous clinical trials are ongoing [67,68].

Recently, other classes of RNA transcripts, such as long non-coding RNAs (lncRNAs) and circular RNAs (circRNAs), gained a lot of attention. lncRNAs are broadly defined as transcripts > 200 nucleotides in length that are transcribed from independent pol II promoters and not translated into protein. lncRNAs comprise a rather heterogeneous class of transcripts that includes intergenic and intronic transcripts, enhancer RNAs, pseudogenes, circular RNAs (circRNAs) and sense and antisense transcripts that overlap with other genes. Currently GENCODE v35 annotates 48,684 lncRNA transcripts from 17,957 lncRNA genes. lncRNA genes can consist of multiple exons, that, upon transcription, are subjected to regular splicing resulting in transcripts that contain a 5′CAP structure and 3′poly (A) tail. The majority of lncRNAs are not highly conserved between species and many lncRNAs display a lineage and/or cancer-specific expression [69]. lncRNAs are found capable of regulating gene expression by diverse mechanisms operating at epigenetic, transcriptional or post-transcriptional levels [68,70,71,72,73]. They either function in *cis*, mediating effects nearby, or in *trans* at distant genomic or cellular locations. lncRNAs have been reported to direct chromatin-modifying complexes to specific gene promoters, to bind transcription factors or RNA-binding proteins, often involved in creating scaffolds facilitating interactions between different biomolecules. They are also known to bind directly to DNA or function as competitive endogeneous RNA (ceRNA) acting as miRNA sponges. Some lncRNAs have been functionally characterized as essential actors in tumorigenesis and tumor maintenance either in oncogenic or tumor-suppressive roles [68,74,75]. However, the relevance and precise functions of the vast majority of lncRNAs and their integration in normal or diseased states remains to be elucidated.

Although the presence of circRNAs was already reported a few decades ago, a publication by Salzman et al. in 2012 renewed the interest in these transcripts by emphasizing their abundance and variety in mammalian cells [76]. CircRNAs are single-stranded, covalently closed circular RNA molecules produced by precursor mRNA back-splicing of exons in which a downstream 5′splice site is linked with an upstream 3′splice site [77]. The process of back-splicing is facilitated by the canonical spliceosomal machinery and regulated by complementary sequences in introns flanking the circularized exons and RNA-binding proteins [78]. It appears circRNAs are found throughout the eukaryotic kingdom and are usually expressed in lineage-specific patterns. Their circular nature endows them with increased stability providing a distinct advantage for use as biomarker. Initially considered as the results of aberrant splicing, it is now recognized that at least some circRNAs fulfil important biological functions [78]. However, so far only few circRNAs have been functionally characterized, a process that is hampered by technical hurdles as circRNAs resemble their linear counterparts [78]. CircRNAs have been implicated in carcinogenesis. Using an exome capture RNA sequencing protocol, a comprehensive catalogue (MiOncoCirc) was generated of circRNAs detected in more than 2000 cancer samples derived from >40 cancer sites including primary and metastatic tumors as well as rare tumor types [79]. MiOncoCirc lists > 125,000 species of cancer-related circRNAs. In general, it is believed that circRNAs can function as ceRNAs capable of sequestering miRNAs and/or RNA-binding proteins [68]. Future research will shed more light on the functional significance of circRNAs in physiological and pathological circumstances and see a further development of their potential as biomarkers.

## 5. Dysregulated miRNAs in GIST

Several research groups examined which miRNAs are aberrantly expressed in GIST as a first step in identifying miRNAs essential for tumorigenesis, maintenance and progression of GIST (see Table 1 for an overview). Subramanian et al. discovered, by analyzing the miRNA expression profiles in various sarcomas, that each sarcoma subtype, including GIST, was characterized by its own unique miRNA expression signature [80]. In addition to *KIT* or *PDGFRA* mutations, GIST displays characteristic genomic alterations, most notably a loss of the long arm of chromosome 14 [81] and deletions of chromosome 1p and 22q [82,83]. Loss of chromosome 14q is seen in approximately 70% of GISTs. Interestingly, Choi et al. reported on the existence of miRNA expression patterns linked to 14q loss. Many miRNAs that are actually located on chromosome 14q appear downregulated [84]. Haller et al. described localization and mutation-dependent miRNA expression patterns in GIST focusing on miR-132, miR-221, miR-222 and miR-504 [85]. Particular attention, by several research groups, has been given to miR-221/222 as these miRNAs were reported to regulate KIT receptor expression [86]. It was shown that miR-221/222 were downregulated in GIST and correlated to KIT expression [87] and also in GIST cells target *KIT* [88,89]. Enhanced expression of miR-222 in GIST cells by miRNA mimics inhibited cell proliferation, affected cell cycle progression and induced apoptosis [88]. Additionally, miR-218 and miR-375-3p were mentioned to regulate *KIT* as well as miR-494, an miRNA associated with 14q loss [90,91,92]. The transient modulation of miR-494 in the GIST882 cell line led to inverse responses in KIT protein levels. Moreover, miR-494 overexpression provoked apoptosis, impaired cellular proliferation and affected the cell cycle. Interestingly, in a subsequent paper, the research group reported that miR-494 also targets survivin (*BIRC5*) [93]. These findings led the authors to propose that miR-494 synergistically suppresses GIST when expressed by targeting both survivin and *KIT*. These *KIT*-targeting miRNAs as well as the *ETV1*-targeting miR-17 and miR-20a [88] may be of therapeutic value, particularly in drug-resistant diseases in which GISTs still rely on KIT signaling. Yamamoto et al. noted that miR-133b was among the downregulated miRNAs in high-grade GIST compared to intermediate and low-grade GISTs and further demonstrated that fascin-1 (*FSCN1*) expression was regulated by miR-133b [94]. It was subsequently shown that an overexpression of FSCN1 correlated to shorter disease-free survival time and aggressive pathological factors. Tong et al. reported miRNA expression profiles that distinguish between malignant and more benign GISTs and between malignant and borderline GISTs [95]. Comparing GISTs with leiomyomas, Fujita et al. described the upregulation of miR-140 in GIST samples [96] but did not indicate potential mRNA targets. The epigenetic silencing of miRNAs in GIST was investigated by Isosaka et al. [97]. An in vitro screen using the cell line GIST-T1 revealed at least 21 miRNAs whose expression was associated with the methylation of an upstream CpG-island. MiR-34a and miR-335, miRNAs found silenced in GIST, were further functionally characterized and were shown to suppress cellular proliferation of GIST-T1 cells when overexpressed. In addition, miR-34a, but not miR-335, affected migratory and invasive processes and was demonstrated to regulate *PDGFRA*. Using novel high-throughput sequencing methods, Gyvyte et al. uncovered and validated miRNAs deregulated in GIST in comparison to adjacent normal tissue [98]. It was found that miR-215-5p levels were negatively correlated with the risk-grade of GIST and that miR-509-3p is upregulated in epitheloid and mixed cell type GIST compared to the spindle type. In a subsequent study the same group focused on miR-200b-3p and miR-375-3p—both were found to be reduced in GIST compared to normal adjacent tissue [91]. These miRNAs negatively affected cell viability and cellular migratory capability when overexpressed in GIST-T1 cells. MiR-200b-3p was demonstrated to directly target *EGFR* and indirectly affected ETV1 protein levels, whereas miR-375-3p targeted *KIT*. A cell line study by Lu et al. revealed that miR-152 is downregulated in GIST cells, and its overexpression inhibited tumor cell proliferation and induced apoptosis [99]. Interestingly, the miR-152 phenotype is mediated through the regulation of cathepsin L (*CTSL*). In search of new miRNA-based treatments for GIST, Long et al. identified the overexpression of miR-374b in GIST and provided evidence that this miRNA targets the tumor suppressor *PTEN* [100]. It is suggested that miR-374b enhances survival, migration and invasion and inhibits apoptosis by stimulating the PI3K/AKT signaling pathway through the downregulation of PTEN. The authors tentatively conclude that inhibition of miR-374b constitutes a novel therapeutic strategy for GIST. A different group highlighted that miR-4510 downregulation, as normally is observed in GIST cells, promotes GIST progression including tumor growth, invasion and metastasis through the increase in apolipoprotein C-II (*APOC2*), shown to be an miR-4510 target [101].

Some studies investigated both mutant GISTs and WT GISTs, describing differentially expressed miRNAs between these GIST subtypes [102,103]. Bioinformatic analyses led Pantaleo et al. to propose the existence of mRNA/miRNA regulatory networks that may be therapeutically targeted in WT GIST [103]. Bachet and coworkers examined miRNA expression profiles in murine NIH3T3 cells expressing either human wild-type *KIT*, hemizygous *KIT* mutants del 557–558 (D6) or del 564–581 (D54), heterozygous *KIT* mutants wild-type/D6 or wild-type/D54 and, for validation purposes, in human GIST samples [104]. Importantly, the authors concluded that miRNA, as well as mRNA, expression profiles depend on the homozygous/heterozygous/hemizygous status of the *KIT* mutations and the deletion/presence of TYR568 and TYR570 residues. These results appear to suggest that different oncogenic pathways are activated and should be further validated using well-characterized GIST samples.

The various screens comparing tumor tissue with adjacent non-cancerous tissue usually indicate many miRNAs that are deregulated in GIST. However, it is often not clear which of the listed miRNAs fulfil a key oncogenic or tumor-suppressive role in cancer-related processes and which miRNAs do not. In-depth, and often time-consuming, functional studies are necessary that should first establish which miRNAs affect cancer-related processes when modulated. Ideally these experiments should be performed both in vitro using well-characterized cell lines and in relevant in vivo models. Once an miRNA is singled out in this way, its target genes and pathways should be identified in the GIST context. To this end, bioinformatics may be used as well as unbiased biochemical approaches, e.g., PAR-CLIP [105]. Once an miRNA target has been defined and validated it is important to verify that the modulation of the target(s), e.g., by RNAi and/or overexpression experiments, phenocopies the miRNA-related cellular phenotype. The findings should be linked to the situation seen in the clinic so some sort of validation using clinical samples is needed to corroborate the clinical relevance. In this respect, much work still needs to be done.

**Table 1 ijms-21-06975-t001:** Dysregulated microRNAs (miRNAs) in Gastrointestinal Stromal Tumors.

miRNAs Up/Downregulated in GIST ^a,c^	Comparison/Number of Samples	Platform	Validated miRNAs; Targets and/or Pathways; Association with Clinicopathological Parameters	Ref.
*Upregulated*: Let-7b; miR-10a; miR-22; miR-29a; **miR-29b**; **miR-29c**; **miR-30a-5p**; miR-30c; miR-30d; miR-30e-5p; miR-99b; miR-125a; miR-140*; miR-143; miR-145 *Downregulated*: miR-1; miR-92; **miR-133a**; **miR-133b**; **miR-200b**; **miR-221**; **miR-222**; **miR-368**; **miR-376a**	Snap-frozen tumor and tissue samples Primary GIST (n = 8) vs. SS (n = 7); LMS (n = 6); DDLPS (n = 1); RMS (n = 6); NSM (n = 5); skeletal muscle (n = 2)	Microarray Sequencing		[80]
*Downregulated*: miR-127; miR-134; miR-136; miR-154; miR-154*; miR-299-5p; miR-299-3p; **miR-323**; miR-329; miR-342; **miR-368**; miR-369-5p; miR-369-3p; **miR-376a**; **miR-376a***; miR-376b; miR-377; miR-379; miR-381; **miR-382**; **miR-409-3p**; **miR-409-5p**; miR-410; miR-411; miR-431; miR-432*; miR-433; miR-485-3p; miR-487a; miR-487b; miR-493-3p; miR-493-5p; **miR-494**; miR-495; miR-539; **miR-625**; miR-654; miR-758	Snap-frozen tumor samples Primary GIST (n = 20) comparing 14q loss (n = 14) vs. 14q presence (n = 6)	Microarray	• Association with 14q loss	[84]
*N.A.*	Snap frozen tumor tissue Discovery: Primary GIST (n = 12) Validation: Primary GIST (n = 49)	Microarray RT-PCR	miR-132; miR-221; miR-222; miR-504 High miR-132 expression level associated with gastric *PDGFRA*-mutated GIST cf. gastric *KIT*-mutated GIST High miR-221 and miR-222 expression levels associated with wild-type GIST cf. GIST with *KIT* or *PDGFRA* mutation High miR-504 expression associated with gastric GIST with *KIT* mutation cf. intestinal GIST with *KIT* mutation	[85]
*Downregulated*: **miR-221**; **miR-222**	FFPE samples Primary GIST and adjacent normal tissue (n = 54 pairs)	RT-PCR	• Association with KIT positivity	[87]
*Downregulated:* **miR-494**	Snap-frozen tumor samples Primary GIST (n = 31)		miR-494; *KIT* • Association with 14 q loss	[84,92]
*Upregulated*: **miR-29c**; **miR-30a**; miR-330-3p; miR-497; miR-603 *Downregulated*: miR-21; **miR-221**; **miR-222**; **miR-382**; miR-938	Snap-frozen tumor samples Primary GIST (n = 50) vs. intestinal LMS (n = 10)	Microarray RT-PCR	miR-17, miR-20a; *ETV1* miR-222; *KIT*	[88]
See publication	FFPE samples Adult *KIT*/*PDGFRA* mutant GIST (n = 30) vs. adult WT GIST (n = 25) vs. pediatric WT GIST (n = 18)	RT-PCR	• Distinct miRNA signatures for GIST subtypes correlating with clinicopathological parameters.	[102]
*Upregulated*: miR-330-3p; miR-455-5p; **miR-455-3p**; miR-886-3p *Downregulated*: miR-129-1-3p; miR-129-5p; miR-214-5p; miR-424; miR-450a; miR-491-5p	Snap-frozen tumor samples Discovery: *KIT*/*PDGFRA* mutant GIST (n = 9) vs. WT GIST (n = 4) Validation: Mutant GIST (n = 13) vs. WT GIST (n = 3)	Microarray RT-PCR	miR-139-5p; miR-148a; miR-193-3p; miR-330-3p; miR-455-5p; miR-129-1-3p; miR-129-2-3p; miR-876-5p miR-139-5p and miR-455-5p predicted to target IGF1R miR-139-5p predicted to target CDK6 miR-330-3p predicted to target CD44	[103]
*Downregulated*: **miR-133b**	Snap-frozen tumor samples Primary GIST (n = 19) comparing high grade vs. intermediate and low grade	Microarray RT-PCR	miR-133b; inverse correlation between fascin-1 and miR-133b • Downregulated in high-grade GIST	[94]
*Downregulated*: miR-218	Snap-frozen tumor and tissue samples, primary GIST (n = 10), normal adjacent tissue (n = 5)	RT-PCR	miR-218; *KIT*	[90]
*Upregulated*: miR-140-3p; miR-483-5p; miR-3151-5p *Downregulated*: miR-28-3p; **miR-133a-3p**; **miR-133b**; miR-195-5p; miR-378f; miR-3135b; miR-4535	Fresh tumor samples Primary GIST (n = 9) vs. leiomyomas (n = 7)	Microarray RT-PCR	miR-140-5p, miR-140-3p	[96]
*Downregulated*: **miR-221**; **miR-222**	FFPE samples Primary GIST (n = 24) vs. smooth muscle (n = 6)	RT-PCR	miR-221/222; *KIT*	[89]
*Downregulated*: miR-9-3p; miR-34a; **miR-152**; miR-155; miR-203; miR-335; **miR-375**; miR-489; miR-582; miR-615; miR-618	GIST-T1 (n = 1), snap-frozen primary GIST samples (n = 39), FFPE primary GIST samples (n = 98)	RT-PCR	miR-34a, *PDGFRA* miR-335 • Association with CpG island methylation	[97]
*Upregulated*: miR-34c-5p; miR-4773 *Downregulated*: Let-7c; miR-218; miR-488*; miR-4683	Snap-frozen tumor samples Primary GIST (n = 53): malignant GIS T (n = 30) vs. benign GIST (n = 9) ^b^	RT-PCR	• Association with malignant GISTs	[95]
*Upregulated*: **miR-196a** *Downregulated*: Let-7c; **miR-29b-2***; **miR-29c***; miR-204; miR-204-3p; miR-218; **miR-625**; miR628-5p; miR-744; miR-891b	Snap-frozen tumor samples Primary GIST (n = 53): malignant GIST (n = 30) vs. borderline GIST (n = 14) ^b^	RT-PCR	• Association with malignant GISTs	[95]
*Upregulated*: **miR-455-3p**; miR-483-5p; miR-509-3p; miR-675-3p *Downregulated*: miR-141-3p; **miR-133a-3p**; **miR-133b**; miR-182-5p; miR-192-5p; miR-200a-3p; **miR-200b-3p**; miR-200c-3p; miR-203a-3p; miR-215-5p; **miR-375**; miR-429; miR-451a; miR-486-5p; miR-490-3p	FFPE tumor and tissue samples Discovery: Pairs (n = 15) primary GIST and adjacent tissue Validation: Pairs (n = 40) primary GIST and adjacent tissue	RNA-seq RT-PCR	All listed miRNAs validated miR-215-5p expression levels are negatively correlated to risk grade miR-509-3p expression levels associated with histological subtype	[98]
*Downregulated*: **miR-152**	Cell lines GIST48; GIST430; GIST882; GIST-T1	RT-PCR	miR-152; *CTSL*	[99]
*Upregulated*: miR-374b	FFPE samples Pairs (n = 143) of Primary GIST and adjacent tissue	RT-PCR	miR-374b; *PTEN* • Association of miR-374b levels with tumor diameter and pathological state	[100]
*Downregulated*: **miR-494**	Snap-frozen tumor samples Primary GIST (n = 35)	Microarray	miR-494; *BIRC5*	[93]
*Upregulated*: **miR-29b-1-5p** *Downregulated*: miR-134-5p; **miR-323b-3p**; **miR-382-5p**; **miR-409-3p**; miR-1185-1-3p; miR-3187-3p; miR-4510	Discovery: Pairs (n = 6) primary GIST and adjacent tissue Validation: Pairs (n = 64) primary GIST and adjacent tissue	RNA-seq RT-PCR	miR-4510; *APOC2* • Association of miR-4510 levels with tumor location, tumor size, mitotic index and risk classification.	[101]
*Downregulated*: **miR-200b-3p**; **miR-375-3p**	FFPE tumor and tissue samples Discovery: Pairs (n = 15) primary GIST and adjacent tissue Validation: Pairs (n = 40) primary GIST and adjacent tissue	RNA-seq RT-PCR	miR-200b-3p; *EGFR* miR-375-3p; *KIT*	[91,98]

^a^ In case multiple miRNAs have been detected only the 10 most significant differentially expressed miRNAs are listed/or miRNAs with the highest fold-change/or miRNAs of which deregulation is validated. ^b^ Classification into benign, borderline and malignant GIST according to [106,107]. ^c^ The miRNAs listed in bold were detected in two of more independent studies.

## 6. MiRNAs Associated with Gastrointestinal Stromal Tumor Metastasis

When GIST metastasizes, treatment becomes more difficult as complete surgical resection is not an option anymore. Biomarkers that indicate whether metastasis is about to occur, or has already occurred, are therefore useful. A limited number and/or small metastatic lesions may be more susceptible to systemic treatment. Several researchers identified miRNAs present in tumors of which the expression levels are associated with metastasis (Table 2). At least 27 miRNAs were found downregulated in high-risk GISTs when 10 high-risk GISTs were compared to four low-risk tumors [84]. Niinuma et al. identified miR-196a as being positively correlated with high-risk grade GIST but also with a poor clinical outcome, tumor size, mitotic count and metastasis [108]. MiR-196a is known to be expressed from the *HOX* gene clusters in mammals. Intriguingly, *HOXC* and the lncRNA HOTAIR were coordinately expressed with miR-196a. MiR-196a inhibition, however, did not affect HOTAIR levels and, conversely, the knockdown of HOTAIR had no effect on miR-196a levels; the authors suggest an epigenetic mechanism underlies the linked expression. In a later paper, the same group demonstrated that the downregulation of miR-186 was observed in tumors that exhibit metastatic recurrence. Analysis of a large validation cohort of 100 primary GISTs uncovered that miR-186 expression is correlated to metastatic recurrence and poor prognosis. It was further shown that the inhibition of miR-186 in a GIST cell line promoted cell migration, most likely by the upregulation of multiple genes implicated in cancer metastasis [109]. Akçakaya et al. identified 44 miRNAs that could distinguish between metastatic and non-metastatic tumors with 19 miRNAs overexpressed and 25 miRNAs underexpressed in metastatic GISTs [110]. Unfortunately, none of these miRNAs were further functionally characterized. MiR-137, an miRNA found downregulated in GIST, was reported to modulate epithelial–mesenchymal transition (EMT) in GIST. Follow-up experiments involving GIST cell lines indicated that miR-137 expression enhanced epithelial cell morphology, possibly by reducing TWIST1 levels. Increased miR-137 levels led to reduced cell migration, activated a G1 cell cycle arrest and induced apoptosis [111]. Similarly, Ding et al. revealed that miR-30c-1-3p, miR-200b-3p and miR-363-3p may modulate EMT and hence invasiveness and consequently metastasis by the regulation of *SNAI2*, a member of the snail C_2_H_2_-type zinc finger transcription factor family [112].

## 7. MiRNAs Related to Imatinib Resistance

Imatinib has been a truly groundbreaking drug for the majority of GIST patients, prolonging overall survival and quality of life [113]. Unfortunately, most GIST patients eventually become insensitive to imatinib and present with a tumor that is progressing and requiring other treatments. Several groups have investigated whether miRNAs can be linked to imatinib resistance (Table 3). These miRNAs can either be used as biomarker signaling drug resistance and possibly tumor progression or alternatively be exploited to obtain an insight into the molecular mechanisms of resistance. Goa et al. compared the miRNA expression profiles of primary imatinib-naïve and imatinib-resistant GIST. MiR-320a, downregulated in imatinib-resistant GIST, was found associated with imatinib resistance although its mode of operation is not further investigated [114]. A cell line study by Fan et al. described that miR-218 is downregulated in resistant GIST contributing to the phenomenon of resistance by regulating PI3K/AKT signaling [115]. Akçakaya and coworkers direct attention to the upregulation of miR-107, miR-125a-5p, miR-134, miR-301a-3p and miR-365 in association with imatinib resistance. A single miRNA, miR-125a-5p, is functionally characterized and shown to regulate *PTPN18* and consequently pFAK levels [110,116]. Zhang et al. performed in silico analyses using GO function and KEGG pathway enrichment as well as the lncRNA–miRNA–target gene regulatory network built of the microarray datasets deposited by Akçakaya et al. [110]. These studies highlighted miR-28-5p and—not surprisingly—miR-125a-5p, both of which displayed a significant correlation to imatinib resistance and imatinib sensitivity [117]. Additionally, Shi et al. uncovered a series of up- or downregulated miRNAs by comparing imatinib-naïve with imatinib-resistant GIST samples [118]. A single miRNA, miR-518a-5p, which is downregulated in imatinib-resistant GIST, was further investigated and demonstrated to bind to the 3′UTR of *PIK3C2A*. It is proposed that the increased PIK3C2A expression affects the cellular response to imatinib and causes resistance. Kou et al. examined the miRNA expression profiles of serum samples derived from GIST patients having an imatinib responsive tumor or a tumor that progresses on the drug [119]. Receiver operating characteristic (ROC) curves demonstrated that miR-518e-5p levels could discriminate serum samples of imatinib-resistant GIST patients from imatinib-sensitive ones with a high sensitivity (99.8%) and specificity (82.1%). Thirty-five differentially expressed miRNAs were detected comparing primary, imatinib-naïve and imatinib-resistant GISTs [30]. An accompanying mRNA profiling of a smaller subset of the same samples uncovered 352 differentially expressed mRNAs; subsequent pathway and network analyses implicated cell cycle and cell proliferation genes as involved in imatinib resistance.

It is noted that the observed differences in miRNA expression between imatinib-sensitive and -resistant GIST tumors are relatively small. Nevertheless, even small miRNA differences may still have a significant impact as diverse miRNAs may act synergistically and the regulation of multiple targets within the same pathway may amplify biological effects [120,121]. Despite a comparable set-up, there is little overlap in imatinib resistance linked miRNAs between the different studies. Of interest in this respect are miR-518a-5p, miR-518e-5p and miR-518d-5p that all derive from a large cluster of miRNAs on chromosome 19q13.42, a chromosomal region that may function in imatinib resistance. However, more extensive research investigating the expression of other miRNA cluster members as well as chromosomal alterations that affect chromosome 19q is needed.

## 8. Additional Non-Coding RNAs in Gastrointestinal Stromal Tumors

Recently the association of lncRNAs with GIST and GIST pathological features was investigated (see Table 4 for an overview). In 2012, Niinuma and coworkers observed that lncRNA HOTAIR expression was associated with high-risk grade GIST, metastasis and poor clinical outcome [108]. The RNAi-mediated knockdown of HOTAIR was shown to inhibit the invasiveness, a surrogate for metastatic potential, of the GIST-T1 cell line. Basically, these findings were confirmed and expanded by others [122,123]. Lee et al. demonstrated that HOTAIR in GIST cells suppressed apoptosis, was associated with cell cycle progression and controlled both invasion and migration [123]. Evidence is presented that HOTAIR through the binding of PRC2 complex components, an epigenetic regulator of gene expression [124], affects the expression of distinct proteins, such as protocadherin 10 (PCDH10), thereby mediating the HOTAIR phenotype. Bure et al. observed that HOTAIR depletion resulted in aberrant DNA methylation patterns through an unknown mechanism, causing either hypo- or hypermethylation patterns that affect gene expression [122]. Hu et al. reported a relative high expression of amine oxidase copper containing 4 pseudogene (AOC4P) in high-risk GIST and noted that also the epithelial–mesenchymal transition (EMT)-related proteins ZEB1, SNAIL and Vimentin were highly expressed [125]. The knockdown of AOC4P affected the migratory and invasive capabilities of GIST cells, induced apoptosis and reduced the EMT. Two reports examined the lncRNA CCDC26 in GIST, indicating its link with imatinib resistance through interacting with KIT and IGF-1R proteins [126,127]. Badalamenti et al. investigated the expression levels of the well-known lncRNAs H19 and MALAT1 in GIST. MALAT1 expression appeared to be associated with KIT’s mutation status. Interestingly, H19 and MALAT1 expression was significantly higher in patients that respond poorly to imatinib i.e., a time-to-progression of <6 months perhaps indicating intrinsic resistance [128]. It is concluded that both H19 and MALAT1 expression levels hold prognostic potential to stratify GIST patients for first-line treatment with imatinib with high expressors indicating a poor response to imatinib. H19 was also detected to be upregulated—together with FENDRR—in GIST samples compared to adjacent normal tissue by Gyvyte et al. [129]. The expression of multiple lncRNAs was analyzed by Yan et al. using a commercially available platform capable of detecting 63,542 lncRNAs and 27,134 mRNAs [130]. Most interestingly, differentially expressed lncRNA and mRNAs between primary GIST and imatinib-resistant GIST were identified. Further, in silico pathway and network analyses implicated the hypoxia-inducible factor 1 pathway as a mediator of imatinib resistance. The role of the lncRNA prostate cancer-associated transcript 6 (PCAT6) was examined by Bai et al. [131]. First, PCAT6 was found to be upregulated in GIST in comparison with adjacent non-cancerous tissue. Follow-up in vitro studies revealed PCAT6 facilitated cancer by repressing apoptosis, enhancing cellular proliferation and—notably—by increasing GIST cell stemness and activating Wnt/β-catenin signaling. Further experiments showed that miR-143-3p is a tumor-suppressive miRNA in GIST as its expression levels are reduced in GIST cell lines in comparison to ICC. An RNA pull-down assay using biotinylated PCAT6 provided evidence that miR-143-3p is sequestered by PCAT6 causing the miR-143-3p target gene peroxiredoxin 5 (*PRDX5*) to be upregulated. Rescue experiments revealed that PCAT6 regulates GIST cell proliferation, apoptosis and stemness by reducing miR-143-3p and enhancing PRDX5.

CircRNAs are a recently recognized class of cellular transcripts that potentially have the capability to affect cellular processes and contribute to pathological processes including cancer [132]. A first study was performed by Jia and coworkers who used ceRNA microarrays that can monitor the expression of 88,371 circRNAs and 18,853 mRNAs [133]. When comparing three pairs of GIST and normal adjacent tissue, a total of 5770 differentially expressed circRNAs and 1815 mRNAs were detected. Three circRNAs (circ_0069765; circ_0084097; circ_0079471) that localized to the host genes *KIT*, *PLAT* and *ETV1* and were upregulated in GIST were further investigated. The circRNAs contained three to six exons of their host genes and their upregulation was confirmed by RT-PCR in a relatively large validation cohort (n = 68). Next, miRNAs predicted to bind to the circRNAs were identified and a circRNA–miRNA–mRNA regulatory network was created. From these studies, the authors concluded that the circRNAs, host genes and miR-142-5p, miR-144-3p and miR-485-3p may be key regulators in GIST.

## 9. Biomarkers

It is evident that non-coding RNAs can be exploited as diagnostic, prognostic and predictive biomarkers. The investigations carried out with GIST report diagnostic miRNA classifiers that distinguish GIST from other sarcomas [80,88,134], identify histological and molecular subtypes [85,98] and define location-specific markers [85]. Of particular clinical interest are the miRNAs associated with relapse risk [84,94,95,98,101,108,109,110,111,112,135] that may be used to predict tumor recurrence and metastasis. These prognostic biomarkers may be further developed into a more quantitative risk evaluation for GIST which is now based on the mitotic index, tumor size and tumor location. Finally, miRNAs associated with imatinib resistance [30,110,114,115,117,118,119] may be used to signal evolving imatinib resistance enabling early clinical intervention. Interestingly, lncRNAs have also been identified that could be used for diagnostic purposes [129,131,133] or are specifically linked to high-risk/advanced GIST [108,122,123,125,128] and imatinib resistance [126,127,130]. However, more research is necessary to select the miRNA classifiers that are most promising for validation in prospective clinical studies. Most studies so far provide proof-of-principle that biomarkers can be identified but do so on a limited number of samples (Table 1). To end up with reliable biomarkers, future studies should avoid caveats and be aware of the critical steps in miRNA-related translational research [136,137,138]. First, appropriately sized sample cohorts should be analyzed taking tumor heterogeneity into account. In addition, the tumor samples must be well-characterized, preferably come from different laboratories and meet certain defined and stringent quality criteria. Ideally, an unbiased, robust and reliable screening procedure should be used that can be standardized and easily executed in different laboratories. For miRNAs, one could consider an RNA-seq approach adapted to suit the class of small RNAs one is interested in. The data should be analyzed using appropriate statistics and the biomarkers should display a defined sensitivity and specificity. It could very well be that a robust classifier needs to be based on the expression of an miRNA panel. For relatively rare tumors such as GISTs—but also for more abundant tumor types—it unavoidable to carry out these studies in international consortia particularly if one intends to bring biomarkers to the clinic [139].

The majority of biomarkers studies on GIST were carried out using tumor samples acquired by invasive biopsies or after tumor resection (Table 1 and Table 3). The exploitation of liquid biopsies, often simple blood draws in a minimally invasive way, have not yet been extensively investigated in GIST patients. Only few investigators examined the miRNA profiles in serum samples. Distinct serum miRNA expression patterns were observed between GIST patients and healthy controls [134] and miR-518e-5p was identified a classifier for imatinib resistance [119]. Circulating miRNAs or other non-coding RNAs—either packaged in extracellular vesicles or not—may signal tumor recurrence, development of drug resistance and tumor progression or indicate metastasis. Particularly, frequent sampling in high-risk patients may indicate disease progression early on, enabling early clinical intervention.

## 10. Therapeutic Potential of Non-Coding RNAs

The mere fact that non-coding RNAs play key roles in carcinogenesis and cancer progression, displaying either oncogenic or tumor-suppressive functions, implies they have therapeutic potential. This has also been demonstrated in various laboratories for GIST. Of interest in this respect are the miRNAs that target the *KIT* receptor: miR-218 [90], miR-221/222 [88,89] and miR-494 [84,92]. Alternative targets with therapeutic potential are *PDGFRA*, reported to be targeted by miR-34a [97], *PTEN* [100], *BIRC5* [93] and *APOC2* [101]. The lncRNAs HOTAIR [108,122,123], AOC4P [125] and PCAT6 [131] are also amenable for therapeutic modulation. In principle, one could restore expression of non-coding RNAs that display reduced levels in cancer using mimics. Conversely, overexpressed non-coding RNAs may be inhibited using antisense approaches. RNA-targeting therapeutic approaches have been discussed in the literature since the discovery of RNAi in the nineties, but encountered significant challenges related to stability, delivery, tissue specificity, tissue penetrance and intracellular trafficking, and toxicities [140,141]. Although many of these issues have not been completely solved, significant advances have been made as exemplified by FDA approved oligonucleotide drugs aimed to induce cleavage of a target mRNA or alter the splicing pattern [142]. RNA oligonucleotides are chemically modified to increase stability, providing protection against nucleases, and improve target binding affinity [143,144]. Most important is the use of 2′-O-methyl substitutions in the sugar backbone of the RNA, 2′-fluoro- or locked nucleic acid (LNA) bases and the use of oligoribonucleotides with phosphorothioate linkages replacing the regular phosphodiester bonds [145]. Moreover, oligonucleotides with a peptide backbone have been generated giving rise to increased stability and binding affinities, and additional modifications, e.g., cholesterol conjugation and cell-penetrating peptides may improve cell uptake. Currently nanoparticles, notably lipid-based nanocarriers and polymer- and peptide particles are being generated. The packaging of oligoribonucleotides in nanoparticles partly overcomes the stability issue and allows for innovative ways to direct the particles to the target tissue [145]. In particular, the progress made in delivery technologies has enabled clinical trials in which non-coding RNA-based therapeutic agents are tested in patients [67,68]. Finally, as the functional significance of the vast majority non-coding RNAs, especially lncRNAs in specific cancers, remains unknown, it is virtually impossible to select the best candidate for therapeutic intervention. This problem may be solved by the use of large-scale CRISPR-CAS9-based screens to rapidly determine the therapeutically actionable lncRNAs [146].

## 11. Future Directions

With the ongoing functional annotation of the non-coding genome comes the realization that non-coding transcripts constitute a central and essential element of eukaryotic biology and, as such, are intimately involved in all kinds of pathological processes including cancer. The clinical relevance of non-coding RNAs is emphasized by many studies listed in the clinical trial database (www.clinicaltrials.gov) that evaluate non-coding RNAs. These trails frequently concern oncological patients in which non-coding RNA expression levels are determined and linked to clinicopathological data for biomarker purposes [68].

For GIST, non-coding RNA biomarkers associated with high-risk GISTs and imatinib resistance may be particularly relevant and obtain a place in the clinical management of this disease. As current biomarker discovery studies are based on relatively small sample cohorts, additional research is required to validate the found biomarker signatures. At the same time the specificity and sensitivity of the biomarker signatures should be determined and how they relate to the traditional clinical and pathological classifiers.

In current clinical practice, advanced GISTs are being effectively treated with imatinib and other small molecule inhibitors targeting the receptor tyrosine kinases KIT and PDGFRA. It seems that, for GIST, the therapeutic targeting of key non-coding RNAs is less relevant. However, eventually all patients develop (multi)drug resistance making the GISTs unresponsive to drugs. In this instance, additional drug targets are needed that, when inhibited or stimulated, affect the ongoing signaling through KIT or PDGFRA signaling. MiRNA or lncRNA-based therapeutic approaches can be of use in this setting. Challenges, however, remain and mainly involve drug safety and targeted delivery issues [142]. Nevertheless, the future will see the enormous potential of the non-coding genome unleashed, revealing new biology followed, undoubtedly, by clinical applications in the form of specific and sensitive biomarkers or the introduction of novel therapeutic strategies.

## Figures and Tables

**Figure 1 ijms-21-06975-f001:**
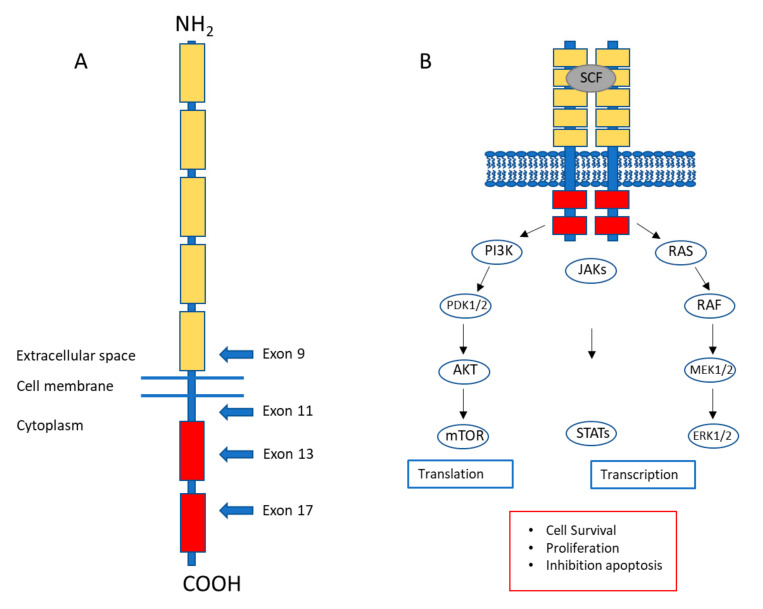
KIT receptor structure and KIT signaling. (**A**) The KIT proto-oncogene codes for a ~110 kDa transmembrane receptor tyrosine kinase KIT (CD117). KIT, together with Platelet Derived Growth Factor Receptor Alpha (PDGFRA), belongs to the type III tyrosine kinase receptor family and consists of 5 extracellular immunoglobulin (Ig)-like domains involved in KIT ligand (Stem Cell Factor, SCF) binding, a transmembrane domain, a juxtamembrane region and an intracellular kinase domain. Mutations in gastrointestinal stromal tumor (GIST) occur in exons that encode functional domains (arrows). (**B**) Constitutive KIT signaling as observed in GIST is transduced through the PI3K/AKT/mTOR, RAS-RAF-MAPK and JAK/STAT pathways thereby inhibiting apoptosis, promoting cell survival and proliferation.

**Table 2 ijms-21-06975-t002:** MicroRNAs Associated with Metastasis in Gastrointestinal Stromal Tumors.

miRNAs ^a,b^	Up/Downregulation	Functional Role	Ref.
miR-146b; **miR-150**; miR-132; miR-342; miR-16; miR-500; miR-212; miR-335; miR-21; miR-199a	Downregulation in high-risk GIST		[84]
**miR-196a**	Upregulation in high-risk GIST		[108]
miR-137	Downregulation in GIST vs. normal adjacent tissue	Regulation of EMT through targeting *TWIST1*	[111]
miR-30c-1-3p; miR-200b-3p; miR-363-3p	Downregulation in SNAI2 high GISTs	Regulation of invasion and migration through targeting *SNAI2*	[112]
miR-186	Downregulation in primary GISTs that exhibit metastatic recurrence	miR-186 is linked to migration and genes implicated in metastasis	[109]
miR-301a-3p **miR-150-3p**; miR-1207-5p; miR-1915	Upregulation in metastatic GIST Downregulation in metastatic GIST		[110]

^a^ In case > 10 miRNAs were identified only the 10 miRNAs with the most significant expression or highest fold-changes are listed. ^b^ The miRNAs listed in bold were detected in two of more independent studies.

**Table 3 ijms-21-06975-t003:** MicroRNAs Associated with Imatinib Resistance in Gastrointestinal Stromal Tumors.

miRNAs Up/Downregulated in Imatinib-Resistant GIST ^a^	Comparison/Number of Samples	Platform	Validated miRNAs; Targets and/or Pathways	Ref.
*Upregulated*: miR-15a; miR-16; miR-151-5p; miR-195 *Downregulated*: miR-140-5p; miR-140-3p; miR-320a; miR-483-5p; miR-574-3p; miR-1280	*Tumor samples* Discovery: primary GIST (imatinib-naïve) (n = 3) vs. Imatinib-resistant GIST(n = 4) Validation: primary GIST (imatinib-naïve) (n = 16) vs. Imatinib-resistant GIST(n = 12)	Microarray RT-PCR	miR-320a	[114]
*Downregulated*: miR-218	*Cell lines* GIST882 vs. GIST430	RT-PCR	miR-218; PI3K/AKT signaling	[115]
*Upregulated*: miR-107; miR-125a-5p; miR-134; miR-301a-3p; miR-365	*Tumor samples* GIST responsive on imatinib (n = 9–16) vs. GIST progressive on imatinib (n = 4–14)	Microarray RT-PCR	miR-125a-5p; *PTPN18* (modulation pFAK levels)	[110,116]
*Upregulated*: miR-491-3p; miR-1260b; miR-2964a-5p; miR-3907 *Downregulated*: miR-221-3p; miR-518a-5p; miR-595; miR-3145-3p; miR-3655; miR-4466	*Tumor samples* Paired (n = 20) primary GIST (imatinib-naïve) vs. Imatinib-resistant GIST	Microarray RT-PCR	miR-518a-5p; *PIK3C2A*	[118]
*Upregulated*: miR-518e-5p; miR-548e	*Serum samples* Imatinib-sensitive GIST(n = 37) vs. Imatinib-resistant GIST(n = 39)	Microarray RT-PCR	miR-518e-5p	[119]
*Upregulated*: miR-28-5p; miR-125a-5p	*Tumor samples* GIST responsive on imatinib vs. GIST progressive on imatinib	In silico analyses of microarray data ^b^	miR-28-5p; miR-125a-5p	[117]
*Upregulated*: miR-92a; miR-118-5p; miR-335; miR-526a/miR-520c-5p/miR-518d-5p; miR-708* *Downregulated*: miR-24; miR-186; miR-455-3p; miR-675; miR-1296	*Tumor samples* GIST imatinib-naïve (n = 33) vs. GIST imatinib-resistant (n = 20)	Microarray RT-PCR		[30]

^a^ In case multiple miRNAs have been detected only the 10 most significant differentially expressed miRNAs are listed. ^b^ miRNA expression data used are from public repository described in Akçakaya et al. (2014) [110].

**Table 4 ijms-21-06975-t004:** Long non-coding RNAs in Gastrointestinal Stromal Tumors.

Lnc RNA	Up/Down Regulation	Functional Role	Ref.
HOTAIR	Upregulation in high-risk GIST cf. low and intermediate GIST	Repression apoptosis Stimulation invasion and migration Stimulation cell proliferation Hypo- and hypermethylation (e.g., PCDH10; DDP4; RASSF1; ALDH1A3)	[108,122,123]
AOC4P	Upregulation in high-risk GIST cf. low and intermediate GIST	Repression apoptosis Stimulation invasion and migration Induction EMT	[125]
CCDC26	Low expression linked to imatinib resistance	CCDC26 interacts with c-KIT and IGF-1R CCDC26 knockdown upregulate c-KIT and IGF-1R	[126,127]
FENDRR, H19	Upregulation in GIST cf. adjacent normal tissue	Positive correlation between H19 and ETV1 Positive correlation between H19 and miR-455-3p	[129]
H19	High expression in advanced GIST with TTP < 6 months		[128]
MALAT1	High expression in advanced GIST with TTP < 6 months	• Correlation with c-KIT mutational status	[128]
TERT-2, OMD-1, ATP7A-2, RERE-4, TCP1-5, FAM108B1-3, C15orf54-4, ATP7A-1 TCF4-6, SNRPN-2	Upregulation in imatinib-resistant GIST Downregulation in imatinib-resistant GIST	• HIF1 pathway regulation	[130]
PCAT6	Upregulation in GIST cf. adjacent normal tissue	Repression apoptosis Stimulation cell proliferation Promotion GIST stemness Activation Wnt/β-catenin signalling Sponging of miR-143-3p	[131]
circ_0069765, circ_0084097, circ_0079471	Upregulation in GIST cf. adjacent normal tissue	• Role in predicted network of circRNAs, host genes (KIT, PLAT, ETV1, resp.) and miR-142-5p, miR-144-3p and 485-3p.	[133]

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
