# Peer review of "Non-Coding RNAs, a Novel Paradigm for the Management of Gastrointestinal Stromal Tumors"

_ijms, 2020, doi:10.3390/ijms21186975_

Round 1

Reviewer 1 Report

Thank you for giving me the opportunity to read this paper. It is interesting review that has the goal to elucidate the role of non-coding RNAs in Gastrointestinal Stromal Tumor. 

I think that the authors have reached their goal. 

I found only few tipying errors (i.e. ANO1 protein at line 33 is written without -, whereas at line 91 is reported as ANO-1).

However, in my opinion this is an interesting review for IJMS readership.

Author Response

Reply: We thank the reviewer for the positive remarks and for pointing out an inconsistency in the ANO1 notation. ANO1 (or DOG1) is the correct designation for this gene, we made sure this is consistently used in the revised manuscript.

Reviewer 2 Report

In my opinion, the overall level of the paper is very good structured: it is well written and several important considerations are highlighted. The discussion sections provide useful information for the readers and the conclusions appear rationale, emphasizing the potential clinical applications of MiRNA or lncRNAs in GIST patients that have developed (multi)drug resistance

Minor comment:

  • references are not complete, there are several recently significant published papers on in this topic

Author Response

Reply: We thank the reviewer for pointing out that we missed literature of relevance for the topic of our review.  Although the reviewer does not mention specific publications, we screened the recent literature and included the following 4 publications in our revised manuscript.

  1. A very recent publication by Gyvyte U. et al. (Int J Mol Sci 2020, 21(14): 5151-) that reports on the functional roles of miR-375-3p and miR-200b-3p both miRNAs that are found downregulated in GIST.
  2. Two publications that profiled WT GIST in addition to mutant GIST by Pantaleo, M.A. et al. (Epigenomics 2016, 8(10): 1347- ) and Kelly, L. et al. (PLOS ONE 2013, 8(5): e64102).
  3. We now also refer to the paper of Bachet, J-B. et al. (PLOS ONE 2013, 8(4): e61103) who describe that gene expression patterns, including miRNA expression patterns, differ depending on the homozygous/heterozygous/hemizygous KIT mutational status and/or the specific nature of some KIT mutations.

We discuss these publications in the text, included them in the tables when appropriate and made sure they appear in the reference list. Please see the specific changes made to the manuscript in the manuscript text that highlights the changes in red. We believe we have sufficiently addressed this minor comment of the reviewer and now present a comprehensive review.
